# Characterizing Soil Profile Salinization in Cotton Fields Using Landsat 8 Time-Series Data in Southern Xinjiang, China

**DOI:** 10.3390/s23157003

**Published:** 2023-08-07

**Authors:** Jiaqiang Wang, Bifeng Hu, Weiyang Liu, Defang Luo, Jie Peng

**Affiliations:** 1College of Agriculture, Tarim University, Alar 843300, China; jiaqiangwang@webmail.hzau.edu.cn (J.W.); 120210019@taru.edu.cn (D.L.); pjzky@taru.edu.cn (J.P.); 2Key Laboratory of Genetic Improvement and Efficient Production for Specialty Crops in Arid Southern Xinjiang of Xinjiang Corps, Tarim University, Alar 843300, China; 3The Research Center of Oasis Agricultural Resources and Environment in Southern Xinjiang, Tarim University, Alar 843300, China; 4Department of Land Resource Management, School of Tourism and Urban Management, Jiangxi University of Finance and Economics, Nanchang 330013, China; hubifeng@zju.edu.cn

**Keywords:** soil salinization, electromagnetic induction, satellite remote sensing, cotton fields, soil profiles

## Abstract

Soil salinization is a major obstacle to land productivity, crop yield and crop quality in arid areas and directly affects food security. Soil profile salt data are key for accurately determining irrigation volumes. To explore the potential for using Landsat 8 time-series data to monitor soil salinization, 172 Landsat 8 images from 2013 to 2019 were obtained from the Alar Reclamation Area of Xinjiang, northwest China. The multiyear extreme dataset was synthesized from the annual maximum or minimum values of 16 vegetation indices, which were combined with the soil conductivity of 540 samples from soil profiles at 0~0.375 m, 0~0.75 m and 0~1.00 m depths in 30 cotton fields with varying degrees of salinization as investigated by EM38-MK2. Three remote sensing monitoring models for soil conductivity at different depths were constructed using the Cubist method, and digital mapping was carried out. The results showed that the Cubist model of soil profile electrical conductivity from 0 to 0.375 m, 0 to 0.75 m and 0 to 1.00 m showed high prediction accuracy, and the determination coefficients of the prediction set were 0.80, 0.74 and 0.72, respectively. Therefore, it is feasible to use a multiyear extreme value for the vegetation index combined with a Cubist modeling method to monitor soil profile salinization at a regional scale.

## 1. Introduction

Soil salinization is a major global problem that threatens land productivity [1]. Severe soil salinization not only causes crop leaf necrosis, crop growth cycle changes and plant death, which eventually lead to decreases in crop yield [2], but also gradually converts fertile, cultivated land to barren land, resulting in a decline in the local economy and an increase in environmental risks [3,4]. Currently, approximately 1.00 × 10^9^ ha of land in hundreds of countries and regions around the world is affected by soluble salts, and salinized soil areas are still increasing at a rate of 2.00 × 10^6^ Ha per year due to desertification and seawater intrusion caused by global warming [5]. China has the largest area of salinized soil in the world, accounting for more than 30% of the world’s salinized soil. There are different types of salinized soils in the northern, northwestern and coastal areas of China [6], especially in Xinjiang, which is located in arid and semiarid areas. The area of salinized soil is approximately 8.5 × 10^4^ km^2^. Accounting for one-third of the saline soil area in China, it ranks first among all provinces in China [7]. With a shift in China’s cultivated land from northeast to northwest [8], Xinjiang has become the largest cotton production base in China, and its cotton output accounted for 87.3% of China’s cotton output in 2020 [9]. However, in recent years, soil salinization in cotton fields has become a major problem, which seriously limits increases in cotton yield and quality. The annual loss caused by soil salinization is approximately $500 million [10,11].

In arid areas, irrigation is usually used to reduce salinity in the soil profile to prevent the salinity in the soil profile from affecting the growth and development of crops. Moreover, the leaching depth of the soil profile varies for different types of crops and for different growth periods of the same crop. For example, the leaching depth of the soil profile in wheat and corn fields is 0–40 cm. The leaching depths of the soil profile in cotton fields are 0–75 cm and 0–100 cm [12]. The irrigation water capacity depends on the salt content of the soil profile. The higher the salt content is, the greater the amount of water needed. The salt data of soil profiles are the main parameters used to calculate irrigation water capacity, but traditional soil profile salt survey methods are insufficient in meeting the needs of regional-scale salt data for soil profiles, and satellite remote sensing technology can yield soil properties quickly, economically and efficiently. Therefore, it is of great significance to explore the potential of satellite remote sensing technology to provide regional-scale soil salt profile information for the precise irrigation of salinized farmland areas.

Satellite remote sensing monitoring models for soil salinization at the regional scale require a certain amount of measured soil profile data. Traditional methods that utilize soil profile excavation or soil drilling consume considerable amounts of manpower, financial resources and time, which increase the cost of satellite remote sensing monitoring and reduce the timeliness of remote sensing monitoring results. Electromagnetic induction technology provides a new way to reduce the cost of soil profile salinization surveys. The apparent conductivity data collected by electromagnetic induction devices such as EM38 and EM31 show good correlations with soil profile salinity [13,14,15]. Only a small number of soil profile samples are collected to build a model between the apparent conductivity and the measured electrical conductivity data, through which the profile electrical conductivity data at all apparent conductivity measurement points can be inverted [16,17,18]. Therefore, when using electromagnetic induction equipment for soil profile salinization information investigation, high-density sampling can be carried out to make the investigation site more representative and improve investigation efficiency. In addition, high-density sampling can improve the degree of agreement between ground survey information and satellite remote sensing information, which is beneficial for improving the precision of remote sensing monitoring model.

In recent decades, with the success of remote sensing monitoring of soil salinization at the local, regional and national scales, many studies have been conducted on a global scale, but few of these studies have investigated desert soil salinization of farmlands by remote sensing monitoring. In particular, few studies have been conducted on the soil salinization of cotton fields under drip irrigation in the oasis of southern Xinjiang, China. Compared with desert soils, farmland soils have higher vegetation coverage during the plant growing season and are more strongly disturbed by human activities, and the degree of soil salinization is relatively low. Generally, farmland soils have low ECe values and high vegetation coverage, so the direct use of spectral information for monitoring soil salinization is limited [19,20]. When crops are subjected to different degrees of salt injury, their photosynthesis rates are affected to different degrees, and the effect is sustained and stable, which is reflected in the canopy spectrum and biomass information [21]. Therefore, remote sensing information from the crop canopy, such as reflectance and NDVI, can be used as monitoring indicators for soil salinization [19]. However, some occasional and transient factors, such as drought, insect pests and field management errors, may also produce characteristics similar to salt damage in crops. To solve this problem, researchers have adopted multiyear or multitemporal remote sensing images to replace remote sensing images from a single period [22,23,24]; this method can also effectively reduce the impact of cloud cover. Cubist modeling method is a machine learning method based on decision tree, which can integrate remote sensing data with other environmental factors, so as to better capture the complex relationship between soil properties and remote sensing data [25]. Meanwhile, Cubist modeling method can also select the most important features for soil property prediction through feature selection of remote sensing data, so as to improve the prediction performance and interpretation of the model.

At present, most studies on remote sensing monitoring of soil salinization focus on the soil surface or the soil layer at specific depth intervals [26]. In general, studies on soil surface salinization are the most common [27,28,29], and only a few researchers have carried out studies on soil layers at multiple depth intervals. Taghizadeh-Mehrjardi et al. [30] used DEM, Landsat 7 ETM+ and ECa data to invert the electrical conductivity of the 0~15, 15~30, 30~60 and 60~100 cm soil layers. The determination coefficients of the model were 0.78, 072, 0.45 and 0.11, respectively. Taghizadeh-Mehrjardi et al. [26] used DEM, Landsat 5 TM and climatic data. The electrical conductivity of the 0~5, 5~15, 15~30, 30~60, 60~100 and 100~200 cm soil layers was inverted by wavelet transform combined with a support vector machine. The correlation coefficients were 0.86, 0.83, 0.83, 0.80, 0.81 and 0.83, respectively. Wang et al. [6] used ECa and DEM data with a random forest (RF) method to construct inversion models of electrical conductivity of the 0–20, 20–40, 40–60, 60–80 and 80–100 cm soil layers, and the determination coefficients of the models were 0.65, 0.55, 0.51, 0.63 and 0.61, respectively. Although the results from a specific soil layer can accurately reflect the distribution of salt in the soil profile at a certain timepoint, due to the action of water and salt migration, the distribution of salt in the soil profile shows strong temporal and spatial variability, and the results cannot provide timely information to achieve precision irrigation. The salt data for the entire section of soil at a certain depth (such as 0–40, 0–75 and 0–100 cm) are relatively more stable and are more timely in guiding precision irrigation.

Different from previous studies, this study was based on modeling and mapping of soil profile salinity, so that the vertical distribution of soil salinity can be understood through remote sensing images. In this study, three soil layers (0–0.375, 0–0.75, 0–1.0 m) from cotton fields in the Alar Reclamation area, a typical oasis in southern Xinjiang, China, where soil salinization is a prominent problem, were selected as the research object. An EM38-MK2 geodetic electrical conductivity meter was used to conduct a field investigation of soil salinization. Based on Landsat 8 remote sensing image data from 2013 to 2019, the maximum and minimum synthetic values of various vegetation indices at different time scales were calculated, and a Cubist model was used to quantitatively invert and map soil electrical conductivity at different profile depths for cotton fields in reclamation areas. The main purpose of this research was to explore the potential for monitoring soil salinization using a synthetic vegetation index with a long time series and to provide a theoretical basis and technical guidance for the precision irrigation of cotton fields and soil salinization improvements in arid and semiarid areas.

## 2. Materials and Methods

### 2.1. Study Area

The Alar Reclamation area is located in southern Xinjiang, China, between 80°30′–81°58′ E and 40°22′–40°57′ N. It is located in the floodplain of the upper reaches of the Tarim River at the confluence of the Aksu River, Hotan River and Yarkant River and is a typical oasis of agricultural areas affected by soil salinization in the Tarim River basin (Figure 1). The Alar reclamation area is 281 km from east to the west and 180 km from north to south. With an altitude of 960–1460 m that gradually increases from southwest to northeast, it has a typical warm, temperate and extreme arid continental desert climate. The annual average precipitation is 40.1–82.5 mm, the annual average evaporation is 1876.6 to 2558.9 mm, and the ratio of evaporation to precipitation is approximately 40. The study area is rich in light and heat resources, with an annual average temperature of 10.7 °C, annual total sunshine duration of 2556.3–2991.8 h, and frost-free period of 185–219 d. In the study area, the land use types include cultivated land, newly cultivated land and desert; the soil texture types are loam, sandy loam and sandy soil; the soil salinity is mainly composed of sulfate, sulfite and salt chloride; the groundwater salinity is 0.6–6 g L^−1^; the groundwater depth is 1–3 m; the soil pH value varies between 7.26–9.23; and the main crops are cotton, wheat, corn and apples. The salinized soil in the study area is widely distributed, causing varying degrees of damage to crops and natural vegetation. Therefore, soil salinization greatly threatens the sustainable development of local agriculture and the ecological environment.

### 2.2. Field Data and Processing

As shown in Figure 2, in this study, soil apparent conductivity data (ECa, mS m^−1^) were collected with EM38-MK2, a geodetic conductivity meter produced by the GEONICS Corporation of Canada. EM38-MK2 provides two measurement modes, EMH and EMV, and each mode can measure the apparent conductivity of two soil profiles at different depths. In EMH mode, the measured depths were 0–0.375 m and 0–0.750 m, and the apparent conductivity data were recorded as EMH0.375 and EMH0.75, respectively. In EMV mode, the measured depths were 0–0.750 m and 0–1.500 m, and the apparent conductivity data were recorded as EMV0.75 and EMV1.5, respectively. Three soil apparent conductivity data (0–0.375 m, 0–0.750 m, 0–1.500 m) at effective depths were obtained.

Soil profile samples were collected using a Rhino S1 soil acquisition system produced by The American Rhino Drilling Company. The Rhino S1 soil acquisition system has a 1 m sampling tube with a built-in hollow PVC core tube with a core diameter of 36 mm. The single tube sampling time was approximately 30–60 s, which completely preserved the undisturbed quality of the soil samples.

Considering the reclamation area size, degree of soil salinization, cotton planting density, traffic conditions and other factors, 30 cotton fields with an area of more than 10 hm^2^ were selected to set up sample plots before cotton was planted in March 2019. According to the spatial resolution of Landsat 8 remote sensing images, the area of each survey quadrat was set to 100 m × 100 m, roughly within the range of 3 pixels × 3 pixels, and soil apparent conductivity data and soil profile samples were collected. The geographical position (latitude and longitude coordinates) of each sample was measured using differential GPS. When the ground sample data were collected, to understand the threshold range of the apparent conductivity data in the plot and to facilitate the selection of soil profile points, 6 continuous lines in EMH and EMV modes parallel to the cotton planting direction and 6 perpendicular to the cotton planting direction were collected. The data volume was approximately 300, the shape was grid-like, and the data spacing was approximately 3 m. When the ground sample data were collected, to understand the threshold range of apparent conductivity data in the plot and to facilitate the selection of soil profile points, continuous linear apparent conductivity data were collected in EMH and EMV modes with 6 lines parallel to the cotton planting direction and 6 lines perpendicular to the cotton planting direction. The volume of data was approximately 300, the shape was grid-like, and the data spacing was approximately 3 m. According to the threshold range observed during the continuous linear apparent conductivity data collection, the apparent conductivity data of 18 feature samples were collected along the continuous linear apparent conductivity data walking route in accordance with the principle of representativeness, and the data interval was greater than 30 m. The EMH and EMV modes of each sample point were measured three times, and the apparent conductivity data obtained were averaged. At the locations where the apparent conductivity data of the 18 feature samples were measured, corresponding soil profile samples with a depth of 1 m were collected using the Rhino S1 soil acquisition system. The bottom and top of the core tubes were sealed with rubber sealing plugs (red at the bottom and black at the top) to prevent water evaporation and salt loss. A total of 540 sets of apparent conductivity data and 540 sets of soil profile samples at different depths were obtained.

After the 540 soil profile samples of whole sections were cut, each tube of soil profile samples was divided into three different depths (0–0.375 m, 0–0.750 m, 0–1.000 m), with a total of 1620 soil samples. After removing residues such as stones, plant roots and mulch, the soil samples were placed on kraft paper to air dry. After air-drying, the air-dried soil samples were ground and passed through a 2 mm sieve for use. Soil samples at each depth were retained at 180 g. After the soil samples were prepared, a filtrate was prepared with a soil–water ratio of 1:1, and soil electrical conductivity was measured with a conductivity meter (DDS-307, Shanghai).

### 2.3. Satellite Image Processing

Remote sensing images were obtained from the United States Geological Survey (USGS) website (https://earthexplorer.usgs.gov/) on February 25th, 2020. In this study, 172 Landsat 8 images were downloaded from the years 2013 to 2019 for the Alar Reclamation Area of southern Xinjiang (row orbit number 146-32), with a spatial resolution of 30 m × 30 m and a transit period of 16 days. The image processing mainly included FLAASH model radiometric calibration, atmospheric correction and image clipping. In addition, 17 vegetation spectral indices were calculated by band math (Table 1). At the same time, the data for the 30 m digital elevation model (DEM) with spatial resolution were downloaded from the USGS website, including 18 remote sensing monitoring modeling factors.

### 2.4. A Soil Electrical Conductivity Inversion Model Was Established Based on Apparent Conductivity Data from Different Profile Depths

To make the soil electrical conductivity at different profile depths corresponding to a single pixel of the Landsat 8 remote sensing image more representative, the random forest (RF) modeling method was used to construct a soil electrical conductivity inversion model at different profile depths based on the apparent conductivity data. Based on the 540 sets of soil samples collected in the study area, a stratified sampling method was used to divide the data into a modeling set and prediction set at a ratio of 2:1. Among them, there were 360 soil samples in the modeling set and 180 soil samples in the prediction set. The results of the RF inversion model at different profile depths are shown in Table 2. The model R^2^ was between 0.79–0.83, and the RPD index of the soil electrical conductivity inversion model at different profile depths shows that the RF model had good prediction capabilities. Continuous linear apparent conductivity data can be inverted to soil electrical conductivity using the RF model. In the same cell, the average value of all soil conductivities represents the soil electrical conductivity value for that cell.

### 2.5. Modeling Method and Accuracy Evaluation

In this study, RF was used to establish a soil electrical conductivity inversion model based on apparent conductivity data [44]. RF is a machine learning method that builds a neural network and uses multiple decision trees to train, classify and predict sample data. Each decision tree unit is unrelated and randomly arranged, which effectively reduces the analysis error of a single classifier. The classification accuracy and model prediction accuracy are improved, and it is suitable for the efficient processing of large-scale data. Its advantages are that it improves classification accuracy and model prediction accuracy, and it can handle data with strong collinearity and noise and data in which the number of variables far exceeds the number of available samples.

The Cubist modeling method was used to construct a soil electrical conductivity inversion model based on the vegetation index of the remote sensing images [45]. The Cubist model is a machine learning method based on predictive modeling tools. The model builds a subset linear prediction model similar to a classification regression tree. Each subset linear prediction model has independent rules, and the condition set in each rule allows different linear models to capture the local linearity of different parts of the predictor space for automatic interactive processing. The regression tree can make the regression tree smaller and achieve higher prediction accuracy [46]. Moreover, the Cubist model can use continuous and categorical predictors that have strong robustness to the predicted value and reveal a more complex decision tree structure [47].

After comprehensively considering the electromagnetic induction inversion model and the remote sensing model, the determination coefficient (R^2^), root mean square error (RMSE), mean absolute error (MAE) and relative percent deviation (RPD) were selected as the commonly used model accuracy evaluation indicators. Among them, R^2^ reflects the correlation strength between the observed value and the predicted value; RMSE tests the predictive ability of the model; MAE represents the average value of the absolute value of the error between the actual value and the predicted value. For RPD, the prediction accuracy was divided into 5 levels; when RPD < 1.5, it indicated that the model could not predict the sample. When 1.5 < RPD < 2, it indicated that the model could roughly estimate the sample. When 2 ≤ RPD < 2.5, it indicated that the model had better predictive capacity for samples. When 2.5 ≤ RPD < 3.0, the model had good predictive capacity. The model had excellent predictive capacity when RPD ≥ 3.0. Models with large R^2^ and RPD and small RMSE and MAE have good predictive ability and high stability [48].

## 3. Results

### 3.1. Statistical Characteristics of Profile Soil Electrical Conductivity

The electromagnetic induction inversion model of soil electrical conductivity at different profile depths inverted the soil profile conductivity data of 30 plots, different depths and a single pixel. The results are shown in Table 3. In the study area, the soil electrical conductivity at different profile depths was between 0.12–26.60 dS m^−1^, the average value was between 2.00–3.90 dS m^−1^, the minimum value was distributed in the 0–0.375 m soil profile, and the maximum value was distributed in the 0–0.375 m soil profile. In the 0–1.000 m soil profile, the conductivity value of the 0–1.000 m soil profile was between 0–0.375 m and 0–1.000 m depth, which indicates that in the study area, even after the cotton field was irrigated in winter, the phenomena of soil salinization surface aggregation, bottom aggregation and central aggregation existed simultaneously. In addition, from the perspective of the coefficient of variation, the coefficient of variation of soil electrical conductivity of each layer was significantly different and decreased with increasing soil depth. The variation coefficient of the 0–0.375 m depth was 112.50%, showing strong variation. The variation coefficients of soil electrical conductivity at 0–0.750 m and 0–1.000 m depths were 83.43% and 79.23%, respectively, showing moderate variation intensities [49], which indicates that the surface soil was more affected by irrigation measures than the deep soil.

### 3.2. Correlation between the Maximum or Minimum Composite Value of the Vegetation Index Based on Time Series and Soil Electrical Conductivity

Using Landsat 8 remote sensing images, the 17 modeling factors COSRI, DVI, EVI, GCI, GDVI, GVI, GLI, GOSAVI, IPVI, LAI, NDVI, GNDVI, NG, NNIR, NR, SAVI and GRVI were calculated for different time scales. The annual maximum or minimum composite value in 2019 and correlation analysis were performed between the 17 modeling factors and the soil profile conductivity at different depths. The GVI and DEM, which had insignificant correlations, were screened out, and the maximum or minimum values of the remaining 16 vegetation indices with extremely significant correlations (*p* < 0.01) for different time series from 2013 to 2019 were synthesized. Among them, NR and NG were negatively correlated with vegetation coverage, and their multiyear minimum values were obtained, while the remaining 14 vegetation indices were positively correlated with vegetation coverage, and their multiyear maximum values were obtained. Avoiding the disturbance of cotton fields by random factors such as stubble, pests and droughts.

Table 4 reveals the optimal time-scale correlation between the conductivity of the soil profiles at different depths and the maximum or minimum values of the modeling index. The results showed that, from the perspective of the synthesis anniversary of the maximum or minimum modeling index, with increasing soil profile depth, the time period corresponding to the highest correlation coefficient between soil electrical conductivity and the extreme value of the modeling factor showed an increasing trend, and the correlation coefficient showed a decreasing trend.

### 3.3. Accuracy and Zoning Rules of the Cubist Model Based on Soil Electrical Conductivity at Different Profile Depths

To further test the effect of applying the Cubist model at different soil profile depths, the 540 groups of soil profile conductivity data at different depths and 16 maximum or minimum composite values of vegetation indices on the optimal time scale from 2013 to 2019 were used as the basis set. After sorting the sample data in ascending order, a stratified sampling method was used to divide them into a modeling set and a prediction set at a ratio of 2:1. Among them, there were 360 sets of sample data in the modeling set and 180 sets of sample rate data in the prediction set. A Cubist remote sensing of different soil profile depths was constructed. The inversion model was monitored, and the results are shown in Table 5.

In the modeling set, the model had the highest accuracy at depths of 0–0.375 m, with an R^2^ of 0.80 and RMSE and MAE values of 0.99 dS m^−1^ and 0.60 dS m^−1^, respectively. The model at a soil profile depth of 0–1.000 m had an R^2^ of 0.73, and the RMSE and MAE were 1.53 dS m^−1^ and 1.18 dS m^−1^, respectively. The model accuracy at depths of 0–0.750 m was slightly lower than that at depths of 0–0.375 m. In the validation set, the R^2^ of the Cubist model with different soil profile depths was 0.80–0.72. Compared with the modeling set, the accuracy of each evaluation index decreased to a certain extent, but the minimum modeling R^2^ value was still above 0.72, indicating that the model was stable.

Table 6 lists the partition rules of the Cubist model under different profile depths and the corresponding linear model for each subregion to intuitively demonstrate the partition of linear subset models of the Cubist model under different profile depths. According to the analysis of the output results of the model, when the Cubist model was divided into regions, the Cubist model at a soil depth of 0–0.375 m had four partitioning rules, and the main controlling factors were NG and LAI. The Cubist model at a depth of 0–0.750 m had seven partitioning rules, and the main control factors were COSRI, NR, IPVI and GDVI. The Cubist model at a depth of 0–1.000 m had five partitioning rules, and the main control factors were NG, NR and GOSAVI.

### 3.4. Analysis of the Spatial Distribution Characteristics of Soil Electrical Conductivity

Hierarchical mapping of salinized soil in cotton fields with different profile depths can not only quantitatively demonstrate the spatial distribution trends in the soil salt profile but also provide scientific guidance for the prevention and control of secondary soil salinization and the rational distribution of water resources in arid areas. According to the soil salinization classification standard [50], the soil electrical conductivity in the study area was divided into five grades. Soil electrical conductivity ranged from 0 to 2 dS m^−1^ for nonsalinized soil, 2 to 4 dS m^−1^ for mildly salinized soil, 4 to 8 dS m^−1^ for moderately salinized soil, 8 to 16 dS m^−1^ for severely salinized soil and >16 dS m^−1^ for salinized soil.

According to the classification standard of the soil salt profile type, the grid layer was classified by a support vector machine. The spatial distribution of soil salinization at different profile depths based on the Cubist model is shown in Figure 3.

Table 7 shows the graded areas (hm^2^) for soil salinization in cotton fields at different profile depths. The total cotton planting area in the study area was approximately 1.08 × 10^5^ hm^2^. Table 6 shows that when the soil profile depth was 0–0.375 m, approximately 92.24% of the cotton planting area was nonsalinized soil, 7.65% was mildly salinized soil, and the proportion of moderate and severe salinized soil was less than 0.1% of the cotton planting area. When the soil profile depth was 0–0.750 m, the proportion of nonsalinized soil in the cotton planting area decreased to 40.19%, and the proportion of mildly and moderately salinized soil significantly increased to 44.70% and 11.99%, respectively. The proportion of severely salinized soil remained essentially unchanged. When the soil profile depth was 0–1.000 m, the proportion of nonsalinized soil continued to decrease to 35.48%, the proportion of mildly and moderately salinized soil further increased to 46.99% and 17.46%, respectively, and the area of severely salinized soil decreased to 0.08%.

In conclusion, after winter irrigation, the salt present in the cotton planting area in the study area was transported to a deep layer in the 0–0.375 m tillage depth, and the soil was essentially nonsalinized. At the 0–0.750 m and 0–1.000 m section depths, the area of nonsalinized soil decreased significantly, while the area of mildly and moderately salinized soil increased significantly. The proportion of severely salinized soil area decreased from 3.12% to 0.08%, but it was still higher than the proportion of severely salinized soil area in the 0–0.375 m depth, showing a general trend toward salt accumulation. At the same time, at a 0–1.000 m depth in the soil profile, the proportion of nonsalinized soil area was still as high as 35.48%, indicating that more than one-third of the soil in the cotton growing area was not affected by soil salinization, and the rest of the cotton growing area could cause different degrees of soil salinization-related damage due to the increasing trend in soil salinity during the cotton growing period.

## 4. Discussion

With the increase in continuous cropping years for cotton in areas in southern Xinjiang, drawing spatial distribution maps of soil salt contents at different profile depths and accurately evaluating soil salinization is key to effectively controlling soil salinization and further improving cotton yields using the limited water resources available in these arid areas. Combining the apparent conductivity data measured by EM38-MK2 and the maximum or minimum composite values of various optimal time scales of the vegetation index and soil electrical conductivity at different depth profiles produces a very effective mapping method to estimate and model soil salinization.

### 4.1. Screening of Modeling Factors

The spectral information contained in a single band is limited, so the combination operation between different bands can be carried out after extracting the spectral reflectance of remote sensing images. Since soil salinization is largely affected by environmental variables such as climate factors, soil physical and chemical properties, topographic factors, spatial location and vegetation factors, the remote sensing data of environmental variables can be introduced to participate in modeling [51]. TRMM precipitation (TRM), evapotranspiration (ET), digital elevation model (DEM) and land surface temperature (LST) products and their combinations are commonly used as environmental variables for soil salinity modeling at different scales [52]. Ma et al. [53] used 10 vegetation index and 10 topographic factors to construct different soil salt estimation models through three different machine learning algorithms, and the results showed that the most important variable was Weighted Difference Vegetation Index (WDVI), followed by DEM. Correlation analysis or regression analysis was carried out between the extracted image vegetation index and the measured soil salt content, and factors with high correlation and high regression coefficient were selected as sensitive bands and sensitive index factors to participate in the modeling, so as to explore the influence degree of different types of factors on the prediction of soil salinization and reduce the redundancy among factors. It is necessary to classify and optimize the modeling factors before participating in modeling [54]. In this study, 18 modeling factors were selected, including 17 vegetation indices and DEM data. After analyzing the correlation between 18 modeling factors and the conductivity of soil profiles at different depths, it was found that the correlation between GVI, DEM and soil electrical conductivity was not significant because GVI is a multiband weighted sum of radiation brightness. The radiation brightness is the result of the integration of atmospheric, solar and environmental radiation and is greatly influenced by external conditions, which is consistent with the results of Elnaggar and Noller [55].

The DEM is an important source of original data for studying and analyzing local topographic features, watersheds and other surface land forms. The study area is close to the northern edge of the Taklimakan Desert and is located in the Tarim Basin, with flat and open terrain. In the cotton growing area, most cotton farmers use laser technology to level the land before planting cotton to prevent the local accumulation of salt, water and trace elements in the soil.

### 4.2. Correlation between Remote Sensing Modeling Factors and Soil Electrical Conductivity at Different Profile Depths

In the 1970s, some researchers calculated the spectral index of the band information carried by remote sensing images and synthesized the annual maximum value for eliminating the influence of cloud cover on imaged data. The research results were mainly applied to land degradation and biomass monitoring in arid areas [56,57,58]. In recent years, the spectral index extremum synthesis methods have been gradually expanded and applied to soil salinization mapping and assessment [59,60,61], but most studies have only focused on the extremum synthesis of single spectral indices, and the research scope is limited to surface soil. As a result, three different sampling depths were set according to the EM38-MK2 measurement range in this study. A Cubist remote sensing monitoring inversion model was constructed based on 16 vegetation indices at different time scales from 2013 to 2019, which effectively prevented interferences from cotton diseases, pest insects, drought and irrigation measures on soil salinity at different profile depths of cotton fields. At the same time, except for the eight vegetation indices, such as COSRI, EVI, GCI, GNDVI, GOSAVI, GRVI, LAI and NR, the time scales corresponding to the correlation coefficients between soil electrical conductivity at different profile depths and the remaining eight vegetation index extreme values were significantly different. With increasing soil profile depth, the corresponding time period also increased, which effectively improved the correlation coefficient between the vegetation index and soil electrical conductivity.

### 4.3. Analysis of Indicator Factors for Soil Salinization in the Study Area

Figure 3 shows the results of the soil salinization classification in cotton planting areas in the study area according to the soil salinization classification standard. The Alar reclamation area is divided into two parts from west to the east by the Tarim River. The scope of human activities has been developing yearly from south to north along the Tarim River; that is, the areas with the longest cotton planting years are distributed on both sides of the Tarim River. In recent years, since the adjustment of the domestic cotton industry structure, the cotton planting area in the Alar reclamation area has gradually expanded to the south and north. Therefore, the soil salinity in the 0–0.375 m profile depth in the study area was affected by the winter irrigation measures applied to the lower part. Salt movement and aggregation led to 92.24% of the areas at this depth showing nonsalinization; depths of 0–0.750 m and 0–1.000 m could clearly observe the cotton growing areas in the north, northwest, south and southeast of the study area. Salinization was more harmful. The main reason was that the newly reclaimed cotton fields and the cotton fields with short planting years were affected by inadequate irrigation and drainage measures and by a shortage in available water resources. The deep soil salinity of the cotton fields could not be effectively discharged. In the following years, due to strong evaporation and sparse rainfall in the arid areas, the bottom soil salinity moved upward, hindering the normal growth and development of cotton and resulting in a decrease in cotton yield or even in failed harvests, which is highly consistent with the results of many field investigations. At the same time, in portions of the southwestern part of the study area near the Tarim River, there were still cotton fields that are more affected by salinization. This is mainly due to the inconsistent soil texture of each cotton field. Some cotton fields with loamy soil texture are due to early salt removal measures. When soil salt moves up and down across the soil profile, a soil compaction layer is gradually formed at a certain depth so that the surface salts cannot effectively move down and be discharged from the cotton field, which makes it difficult to improve soil salinization in these cotton fields in later stages. The limitations of this study are mainly reflected in two aspects. On the one hand, due to the differences of crops, farming practices, climatic conditions and other factors, the estimation model of soil profile salinity established in this study was only for cotton fields and could not be extended to other fields. On the other hand, due to the wide variety of remote sensing images, the model in this study is only applicable to OLI images of Landsat 8 but not to other satellite images

## 5. Conclusions

In this study, a RF model was used to build a soil electrical conductivity inversion model based on apparent conductivity data for soil salinization monitoring at different profile depths in an oasis of agricultural cotton fields in an arid region. The results showed that the validation sets had R^2^ values of 0.79, 0.83 and 0.81 at 0–0.375 m, 0–0.750 m and 0–1.000 m, respectively; the RMSE was 0.64 to 0.91 dS m^−1^, MAE was 0.36 to 0.54 dS m^−1^, and the RPD was 2.0 to 2.5. The model inverted the soil electrical conductivity data well and made the soil electrical conductivity of different profile depths corresponding to a single pixel more representative. When a Cubist model was used to construct soil electrical conductivity inversion models based on the maximum or minimum values of various vegetation indices at the optimal time scale, the R^2^ values of the validation sets for different depth profiles were 0.80, 0.74 and 0.72, respectively. The RMSE ranged from 0.86 to 0.91 dS m^−1^. The MAE ranged from 0.55 to 1.08 dS m^−1^. Based on the results of the model evaluation indices, the Cubist model could predict soil electrical conductivity at the different depth profiles. The results showed that the farmlands that were severely affected by soil salinization were mainly distributed in the north, northwest, south and southeast of the cotton growing area in the study area, and soil salt showed distinct bottom-aggregation distribution characteristics at the 0–0.750 m and 0–1.000 m depths. These outcomes provide an effective solution for the contradiction between agricultural water consumption and farmland soil salinization in arid areas. Future research will mainly focus on synergistic application of multi-temporal and multi-sensor satellite data resources in remote sensing monitoring of soil salinization in farmland and further exploration of soil salinization monitoring of other staple crops or cash crops, so as to further promote the development of precision and wisdom in the domestic agricultural field.

## Figures and Tables

**Figure 1 sensors-23-07003-f001:**
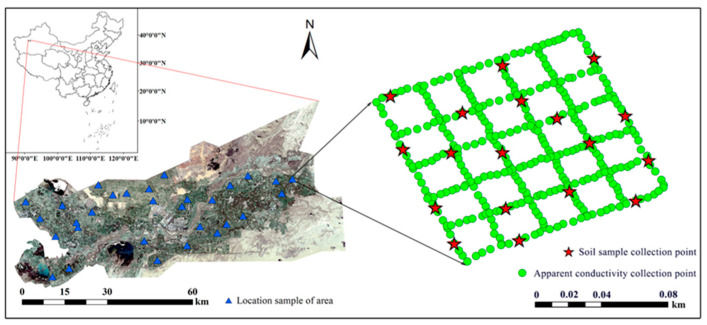
The geographical location of the study area and the distribution of survey samples.

**Figure 2 sensors-23-07003-f002:**
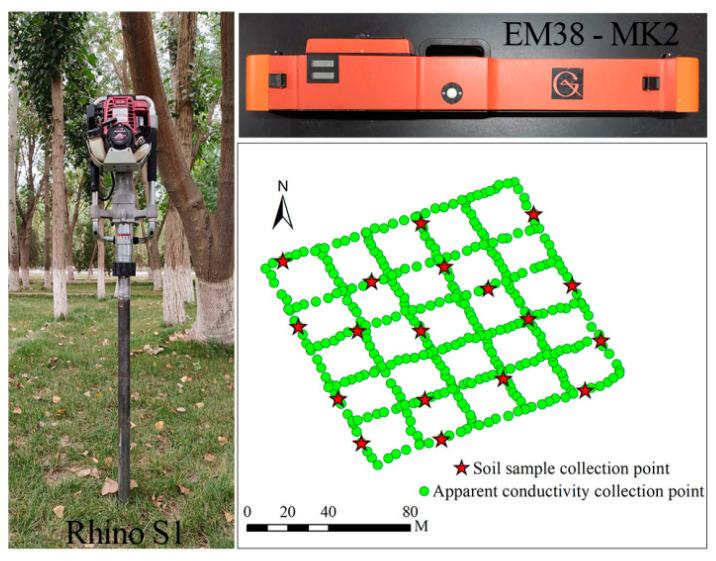
Distribution of EM38-MK2 measuring points and soil sample collection points.

**Figure 3 sensors-23-07003-f003:**
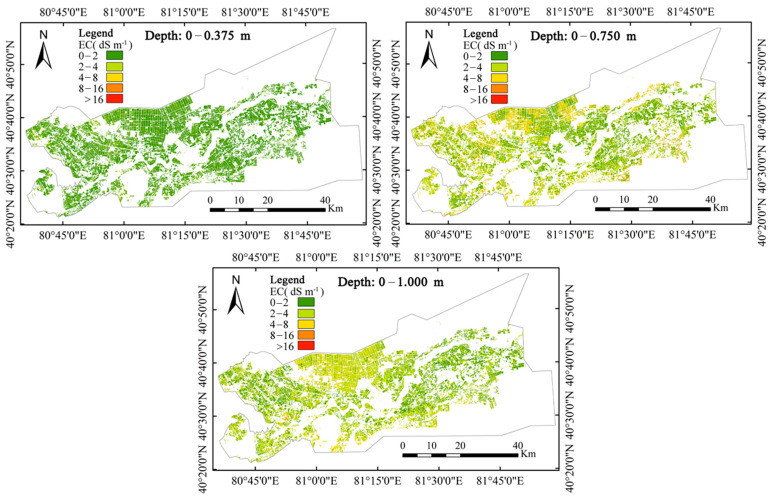
Distribution of soil salinization at different soil profile depths.

**Table 1 sensors-23-07003-t001:** Vegetation index used in this study.

Vegetation Index	Computational Formula	References
Normalized Difference Vegetation Index (NDVI)	(NIR − R)/(NIR + R)	[31]
Green Normalized Difference Vegetation Index (GNDVI)	(NIR − G)/(NIR + G)	[32]
Green Ratio Vegetation Index (GRVI)	NIR/G	[33]
Enhanced Vegetation Index (EVI)	2.5 × (NIR − R)/(NIR + 6 × R − 7.5 × B + 1)	[34]
Difference Vegetation index (DVI)	NIR − R	[35]
Green Difference Vegetation Index (GDVI)	NIR − G	[35]
Green Vegetation Index (GVI)	0.2848 × B − 0.2435 × G + 0.5436 × R + 0.7243 × NIR + 0.084 × SWIR1 − 0.18 × SWIR2	[36]
Leaf Area Index (LAI)	3.618 × [2.5 × (NIR − R)/(NIR + 6 × R − 7.5 × B + 1)] − 0.118	[37]
Soil Adjusted Vegetation Index (SAVI)	1.5 × (NIR − R)/(NIR + R + 0.5)	[38]
Green Optimized Soil Adjusted Vegetation Index (GOSAVI)	(NIR − G)/(NIR + G + 0.16)	[38]
Green Normalized Index (GNI)	G/(NIR + R + G)	[39]
Red Normalized Index (RNI)	R/(NIR + R + G)	[39]
NIR Normalized Index (NNI)	NIR/(NIR + R + G)	[39]
Infrared Percentage Vegetation Index (IPVI)	NIR/(NIR + R)	[40]
Green Leaf Index (GLI)	[(G − R) + (G − B)]/(2 × G) + R + B	[41]
Composite Spectral Response Index (COSRI)	(B + G)/(R + NIR) × (NIR − R)/(NIR + R)	[42]
Green Chlorophyll Index (GCI)	(NIR/G) − 1	[43]

Abbreviation: B, G, R and NIR represent the blue, green, red and near-infrared bands, respectively; SWIR1 and SWIR2 represent the first and second shortwave infrared bands, respectively.

**Table 2 sensors-23-07003-t002:** Accuracy of the soil conductivity model at different profile depths.

Soil Depth (m)	Modeling			Validation			
	R^2^	RMSE (dS m^−1^)	MAE (dS m^−1^)	R^2^	RMSE (dS m^−1^)	MAE (dS m^−1^)	RPD
0–0.375	0.80	0.61	0.59	0.79	0.64	0.36	2.13
0–0.750	0.85	0.78	0.76	0.83	0.86	0.45	2.47
0–1.000	0.84	0.90	0.82	0.81	0.91	0.54	2.34

**Table 3 sensors-23-07003-t003:** Statistical feature values of electrical conductivity in profile.

Soil Depth (m)	n	Minimum (dS m^−1^)	Maximum (dS m^−1^)	Mean (dS m^−1^)	Standard Deviation (dS m^−1^)	C.V. (%)
0–0.375	540	0.12	21.85	2.00	2.25	112.50%
0–0.750	540	0.28	22.00	3.44	2.87	83.43%
0–1.000	540	0.25	26.60	3.90	3.09	79.23%

**Table 4 sensors-23-07003-t004:** Optimum temporal scale correlation between conductivity of soil profiles at different depths and the modeling factor.

Factors	Time Anniversary (Year)			Correlation Coefficient		
	0–0.375 m	0–0.750 m	0–1.000 m	0–0.375 m	0–0.750 m	0–1.000 m
COSRI	1	1	1	−0.33 **	−0.29 **	−0.28 **
DVI	2	5	5	−0.47 **	−0.42 **	−0.42 **
EVI	1	1	1	−0.50 **	−0.43 **	−0.40 **
GCI	1	1	1	−0.47 **	−0.39 **	−0.35 **
GDVI	2	7	7	−0.46 **	−0.41 **	−0.40 **
GLI	1	2	4	−0.46 **	−0.45 **	−0.43 **
GNDVI	1	1	1	−0.53 **	−0.44 **	−0.42 **
GOSAVI	1	1	1	−0.53 **	−0.45 **	−0.41 **
GRVI	1	1	1	−0.47 **	−0.39 **	−0.35 **
IPVI	1	1	2	−0.51 **	−0.46 **	−0.45 **
LAI	1	1	1	−0.50 **	−0.43 **	−0.40 **
NDVI	1	1	2	−0.51 **	−0.46 **	−0.45 **
NG	2	2	7	0.55 **	0.44 **	0.43 **
NNIR	1	1	2	−0.52 **	−0.45 **	−0.44 **
NR	1	1	1	0.54 **	0.49 **	0.46 **
SAVI	1	1	2	−0.51 **	−0.45 **	−0.42 **

** Significant at the 0.01 probability level.

**Table 5 sensors-23-07003-t005:** Accuracy comparison of Cubist models based on different depths.

Soil Depth (m)	Modeling			Validation			
	R^2^	RMSE (dS m^−1^)	MAE (dS m^−1^)	R^2^	RMSE (dS m^−1^)	MAE (dS m^−1^)	RPD
0–0.375	0.81	0.99	0.60	0.80	0.86	0.55	2.28
0–0.750	0.79	1.33	0.99	0.74	1.50	1.08	1.94
0–1.000	0.73	1.53	1.26	0.72	1.35	0.98	1.83

**Table 6 sensors-23-07003-t006:** Classification rules and corresponding models based on the Cubist modeling method.

Soil Depth (m)	Regions	Rules	Models
0–0.375	1	NG ≤ 0.143786	EC = 8.05942 − 29.2 SAVI + 18.2 EVI − 3.4 DVI − 1.4 IPVI
	2	LAI > 1.21532	EC = −31.8621 + 380 NG − 10.13 LAI
		NG > 0.143786	
		NG ≤ 0.174369	
	3	LAI > 1.21532	EC = 83.5739 − 26.29 LAI − 181 NG
		NG > 0.174369	
	4	LAI ≤ 1.21532	EC = 7.35965 − 26.2 SAVI + 18.2 EVI − 3.1 DVI
0–0.750	1	NR ≤ 0.0645848	EC = 0.42251 + 0.17 GCI
		COSRI > 0.1897921	
	2	NR > 0.192525	EC = 5.41577 − 410.1 NR + 479 COSRI
		COSRI > 0.199917	
	3	GDVI > 0.4129	EC = − 17.9032 − 83.9 NNIR + 101.7 IPVI − 82.8 GDVI + 194 COSRI − 1.4 GNDVI + 1.5 GOSAVI − 0.8 SAVI
		IPVI ≤ 0.918396	
		COSRI ≤ 0.1897921	
	4	IPVI > 0.918396	EC = 45.5945 − 63.1 GDVI − 80.3 NR − 109 COSRI + 14.4 SAVI + 4.3 DVI + 4.1 GOSAVI
		COSRI ≤ 0.1897921	
	5	GDVI ≤ 0.4129	EC = −12.7175 + 66.7 NR − 25.4 GDVI + 22.2 DVI + 18.1 GOSAVI
		NR <= 0.192525	
		COSRI ≤ 0.1897921	
	6	NR > 0.0645848	EC = −18.4668 + 28.9 SAVI + 56 NR
		COSRI > 0.1897921	
	7	NR > 0.192525	EC = 5.705
		COSRI ≤ 0.199917	
0–1.000	1	NG ≤ 0.230297	EC = 55.7647 − 139.1 NR + 103 NNIR − 140 IPVI − 9.8 GLI
		NR ≤ 0.062556	
	2	NG > 0.230297	EC = 79.8965 − 82.7 NNIR
	3	GOSAVI ≤ 0.5833951	EC = 8.8938 + 37.4 NR − 22.9 NG − 5.8 IPVI
		NG > 0.10685	
		NG ≤ 0.230297	
	4	GOSAVI > 0.5833951	EC = 34.9597 − 44.8 SAVI
		NG > 0.10685	
		NR > 0.062556	
	5	NG ≤ 0.10685	EC = −7.5859 − 42.7 GLI + 33.7 SAVI
		NR > 0.062556	

**Table 7 sensors-23-07003-t007:** Graded area of salinized soil at different depths of the cotton field profile (hm^2^).

Soil Depth (m)	Soil Conductivity Classification (dS m^−1^)				
	0–2	2–4	4–8	8–16	>16
0–0.375	99,515.16	8256.42	89.28	21.24	1.35
0–0.750	43,358.85	48,224.52	12,938.22	3360.69	1.17
0–1.000	38,275.29	50,692.05	18,831.87	84.15	0.09

## Data Availability

The datasets used in the current study are available from the corresponding author upon request.

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
