# Peer review of "Characterizing Soil Profile Salinization in Cotton Fields Using Landsat 8 Time-Series Data in Southern Xinjiang, China"

_sensors, 2023, doi:10.3390/s23157003_

Round 1

Reviewer 1 Report

The salt data of soil profiles are the main parameters used to calculate irrigation water capacity, but traditional soil profile salt survey methods are insufficient in meeting the needs of regional-scale salt data for soil profiles, and satellite remote sensing technology can yield soil properties quickly, economically and efficiently.

Abstract

In general, the abstract is not complete and explicit enough. The abstract should stand alone and reflect all and major parts of the study. The following key parts of the abstract did not mention 1- The sampling method, modeling method.

Introduction

“The leaching depths of the soil profile in cotton fields are 0-75 cm and 0-100 cm.” This sentence needs to be referenced to show where it came from.

I have seen that the whole Introduction focuses on RS and soil salinity. RS data could be used to estimate a wide range of soil properties. It is advisable to add one paragraph to demonstrate the advantage of Cubist modeling method with a combination of RS data to estimate and predict soil attributes.

Materials and Methods

Why choose the depth of soil profile 0~0.375m, 0~0.750m and 0~1.500m? Please explain the reasonability.

Discussion

The discussion on modeling factors is not deep enough, so it is necessary to compare the current modeling factors on soil salt inversion with the modeling factors in this study to explain the rationality of the modeling factors selected in this study.

The salt data of soil profiles are the main parameters used to calculate irrigation water capacity, but traditional soil profile salt survey methods are insufficient in meeting the needs of regional-scale salt data for soil profiles, and satellite remote sensing technology can yield soil properties quickly, economically and efficiently.

Abstract

In general, the abstract is not complete and explicit enough. The abstract should stand alone and reflect all and major parts of the study. The following key parts of the abstract did not mention 1- The sampling method, modeling method.

Introduction

“The leaching depths of the soil profile in cotton fields are 0-75 cm and 0-100 cm.” This sentence needs to be referenced to show where it came from.

I have seen that the whole Introduction focuses on RS and soil salinity. RS data could be used to estimate a wide range of soil properties. It is advisable to add one paragraph to demonstrate the advantage of Cubist modeling method with a combination of RS data to estimate and predict soil attributes.

Materials and Methods

Why choose the depth of soil profile 0~0.375m, 0~0.750m and 0~1.500m? Please explain the reasonability.

Discussion

The discussion on modeling factors is not deep enough, so it is necessary to compare the current modeling factors on soil salt inversion with the modeling factors in this study to explain the rationality of the modeling factors selected in this study.

Reviewer 2 Report

Comments are attached.

Round 2

Reviewer 2 Report

It is recommended for publication in the Sensors journal.